

# Use *ATCCfinder* to identify commercially available American Type Culture Collection strains based on sequence queries

Samuel I. Koehler, Earl A. Middlebrook, Blake T. Hovde and Erik R. Hanschen

Bioscience Division, Los Alamos National Laboratory, Los Alamos, NM, United States of America

Corresponding authors
Samuel I. Koehler, sikoehler@lanl.gov
Erik R. Hanschen, hanschen@lanl.gov

## ABSTRACT

Microbiology research was conducted for decades before widespread availability of sequencing resources and large culture collection sequence repositories, making it challenging to efficiently identify and validate strains used in historical studies. Similarly, finding commercially available microbe strains similar to strains of interest, or containing target genes of interest found during metagenomic experiments is challenging. Despite tremendous advances in sequencing data availability, database curation, and sequence-searching software capabilities, identifying commercially available microbe strains from sequence data remains complicated and tedious. The American Type Culture Collection (ATCC) is an organization selling a wide variety of microbes, uniquely providing strain-level taxonomy classification and associated sequenced reference genomes for over four thousand isolates, with more being added regularly. As researchers purchase and sequence isolates from ATCC, many sequences derived from ATCC isolates are deposited on public databases such as NCBI-Genome. Sequences uploaded to public databases will vary in laboratory, bioinformatics, and metadata quality and can also contain mutations derived from cultivation which are not representative of ATCC stocks. Using ATCC-sourced reference genomes ensures consistent quality and analysis methodologies are implemented to accurately represent strain sequences. Currently, ATCC does not provide methods to search for sequence similarity between many query sequences and ATCC genomes. While NCBI-BLAST could be used to search for queries against GenBank, with results filtered for "ATCC" tags, search result quality varies and requires time-consuming sorting. Here we present the software *ATCCfinder* (GitHub: https://github.com/lanl/ATCCfinder, Zenodo: https://doi.org/10.5281/zenodo.15178103), utilizing ATCC application interface software (API) to generate query-able databases from ATCC genome resources. The algorithm generates databases of the four ATCC data types: strain-specific genome assembly sequence data (sequence), information about how each strain was collected (metadata, catalogue), and structural/functional information about genome assemblies (annotation). Once ATCC sequences are retrieved by *ATCCfinder*, nucleotide queries are compared against ATCC reference genomes *via* sequence alignment tool minimap2, with results parsed and analyzed to produce summaries describing ATCC-available strain homologous sequence matches. *ATCCfinder* identifies and downloads new ATCC references, allowing users to maintain an updated target search database. *ATCCfinder* efficiently accesses, queries, and summarizes ATCC resources, identifying

purchasable strains homologous to historical sequences, functional genes, operons, and other genetic components.

## INTRODUCTION

A core component of microbial research is experimenting with cultured organisms in the laboratory. Genomics enables researchers to define and identify species by genome sequence, enabling scientists to select suitable organisms for studying biological systems of interest, such as genes within a metabolic pathway. Researchers use sequence alignment homology to identify species containing target systems, then culture and study these species in the lab. If genomic elements are identified through metagenomic experiments, published sequence data, or from publications prior to the genomic era, identifying candidates to order that harbor these sequences can be a complicated and tedious task. Many powerful tools exist for each step within the sequence-to-procurement workflow, however the diversity of available databases, variability of strains available for culturing, and differences in analysis objectives result in no existing tools for performing this analysis automatically. A major issue is finding strains available for culturing. An achievement in genomics is the development of massive databases containing high-quality sequence information such as GenBank (*Benson et al., 2015*), GEO (*Clough et al., 2024*), and UNIPROT (*The UniProt Consortium, 2021*). Sequence databases link nucleotide sequences with species classification data so that sequence homology alignments identify taxa that can be found as strains for procurement. However, major sequence repositories do not link classification information to procurable strains, meaning substantial time associated with sequence-to-culturing work is spent researching whether or not a strain is available for procurement. Alternatively, many retailers sell microbial strains, notably the American Type Culture Collection (ATCC) (*Nguyen et al., 2024*), DSMZ (*Leibniz Institute, 2024*), the Microbial Culture Collection (*Sharma & Shouche, 2014*), the ARS Culture Collection (NRRL) (*The USDA-ARS Culture Collection, 2024*), and the National Collection of Type Cultures (NCTC) (*The National Collection of Type Cultures, 2024*). So far, only ATCC directly offers supplementary sequence information to enable sequence searches yielding homology alignments directly associated with purchasable strains. The software MicrobeDB (*Langille et al., 2012*) offers tools for downloading and organizing thousands of genomes for analysis, which could be used to curate species sets with sequenced genomes and known procurement sources, however this approach requires frequent upkeep since quality and availability varies dramatically across reference sequences. Searching gene or amplicon queries against target databases listed above is performed online *via* integrated BLAST (*Camacho et al., 2009*) interfaces. Once matches have been identified, external culture databases need consulting to confirm strain procurement, however, there is no guarantee that the metadata linking the BLAST hit to the culture collection is accurate and that the sequence data represents

the stock culture. A tool is needed that automates sequence-to-culturing workflows and enables researchers to query sequences against target databases of high-quality, consistent reference sequences with pre-verified procurement availability.

Despite the dramatic increase in metagenomic experiments, there are no standard methods for taking sequences such as a single gene, many genes in a pathway, or a primer sequence, and finding a commercially available organism with homologous sequences. Common workflows involve complex steps and specialized bioinformatics skills: implementing a mixture of accessing and downloading reference genomes from public databases such as NCBI-RefSeq (*O'Leary et al., 2016*) followed by sequence homology searching *via* bioinformatics alignment software like BLAST (*Camacho et al., 2009*) and manual assessment of strain availability. Given a sequence of interest, whether from an older publication with limited sequencing data and associated culture collection strain, from a current research interest pursuing additional bacterial strains, or a metagenomic experiment that couldn't culture an organism of interest, one begins by identifying a suitable database to query against. Relevant databases are massive, requiring significant time and computational resources to download, store, and keep updated. Some databases avoid requiring local download by directly offering sequence searching capability built into their web interfaces, but running sequence alignments online requires waiting in queues and limited query amounts (*The UniProt Consortium, 2021*; *O'Leary et al., 2016*; *Yarmosh et al., 2022*). Once sequence alignments are generated against a database, one filters results by criteria including the alignment quality (alignment length and number of matching bases, gaps and mismatches), the quantity of total alignments, and the taxonomic breadth of the reference database. Because different databases are composed of different qualities and quantities of samples and species, filtering criteria is reevaluated for each database scenario and query sequence. Once the filtering scheme is optimized and high-quality alignments have been identified, one must consider confidence of classification before proceeding to procuring microbes: were there multiple high-quality hits to different species? Did the database queried against produce any convincing alignments? Did the best alignment only cover a third of the queried sequence, suggesting that classification results may identify the correct genus but not exact species? Was the best alignment biased by a short but high percent identity match when a different strain has much longer alignments with slightly lower percent identity? Are hits dominated by one over represented taxa, masking a less studied one of interest? Finally, if the filtered alignments suggest that a sequence belongs to a specific species, the microbe is validated as available for purchase and is suitable for culturing, which is often false. In this situation, one continues down the list of filtered alignments belonging to a specific species of suitable microbe that is commercially available with no guarantee that the related taxa will harbor the sequence of interest. Researchers query databases for many reasons, not all of which require summary information aggregating alignment results across target sequences. For example, databases such as NCBI RefSeq and UniProt are frequently searched against to find top sequence homology matches for the purposes of sequence identification. However, for scientists interested in finding organisms containing a homologous sequence for culturing and experimentation, getting summary information about all potential

homologous matches is highly valuable but requires much manual filtering and data processing to complete. *ATCCfinder* was designed to streamline the tedious process of sequence-to-strain identification and purchasing. Using ATCC Genome Portal references, we ensure that any target species producing high-quality alignments is available for purchase. Additionally, the ATCC Genome Portal currently contains over 4,000 reference genomes that, when compressed, comprise ~9 GB storage space. As the ATCC Genome Portal expands its database size, it will remain a manageable size for local download for many years to come. Alignment methods and filtering criteria are standardized, creating confident and reusable results across analysis queries. Minimal dependencies means that *ATCCfinder* is easily downloadable and deployable in minutes. It is notable that the ATCC Genome Portal contains a high-quality web tool (Sequence Search) which allows users to search a query against reference genomes for sequence homology. While the authors recommend this tool for single-search scenarios, it only allows one query at a time, provides minimal alignment statistics, and does not have available options for downloading alignment summary information as a data table. *ATCCfinder* was created to provide nuanced, automated analysis and reporting of multiple query sequences.

*ATCCfinder* is validated using diverse sequence types to ensure that the tool has broad applicability and high species identification efficacy regardless of genera, sequence length, or sequence type. Four types of query sequence were used for *ATCCfinder*'s validation. (1) Bacterial 16S ribosomal RNA (rRNA) and fungal Internal Transcribed Spacer (ITS) sequences are often relatively short (<2 kbp), ubiquitous, and highly conserved due to their essential functioning within ribosomes, making them reliable sequencing data targets for performing metagenomic taxonomic classification. (2) Coding sequences (CDS) are the regions of genes translated into functional protein sequences and are approximately captured by RNA-Seq experiments performed across the tree of life to analyze a species' transcriptional response to an experimental condition. While the previous two sequencing data sources, rRNA and CDS, capture major swaths of sequences one may expect to encounter and query within the tool, (3) an additional category of *randomly* selected genomic sequences was used to account for events when a sequence of interest may not have known or annotated function. (4) Actual sequence data generated by extraction and sequencing of a known microbial community sample validates *ATCCfinder* with short-read data. By considering diverse sequence types, we demonstrate that a majority of representative sequence queries were accurately estimated. *ATCCfinder* downloads available reference sequences, generates an up-to-date database, and accurately identifies available ATCC culture strains based on a variety of sequence queries.

# MATERIALS & METHODS

## Overview

*ATCCfinder* enables researchers to query ATCC reference genome assemblies, identifying candidate species with sequence homology for future experimentation. Testing the software against many sequence types ensures *ATCCfinder* accurately reports and summarizes sequence matches across diverse inputs. The following sections describe the *ATCCfinder*

algorithm for downloading and querying reference sequences (*ATCCfinder* algorithms), ATCC reference retrieval for use as the target database (ATCC genome retrieval), diverse query sequences curated for testing of *ATCCfinder* functionality (Algorithm validation using a known microbial community sample), and testing *ATCCfinder* using real sequencing reads generated in our lab (Microbial community DNA extraction, sequencing, & processing).

### *ATCCfinder* algorithms

*ATCCfinder* is composed of two core scripts for the purposes of downloading and searching against ATCC reference sequences: 'download.py' and 'search.R' written in Python (*Van Rossum & Drake, 1995*) and R (*R Core Team, 2022*) respectively. The download.py algorithm is a wrapper for the ATCC software *genome_portal_api* (*ATCC-Bioinformatics, 2023*) and utilizes packages argparse, gzip, os, re, and time to communicate with ATCC servers retrieving ATCC databases of genome sequences, metadata, annotations, and catalogue information (Fig. 1). Most public databases integrate BLAST to perform alignment. Here, we use minimap2 (*Li, 2018*) for its speed and ability to efficiently handle long sequences. The search.R tool uses minimap2 (default parameters = "-c", "-x map-ont", '—paf-no-hit) to align user-defined nucleic acid query sequences against a reference data set of FASTA-formatted files. The resulting pairwise mapping format (PAF) alignment file is parsed to summarize query alignment results, with metrics 'alignment score' (AS) and 'nmatch' maximum values used to determine best target sequence matches (Fig. 2). Target alignments with the highest combined score are quantified and taxonomically described in a summary output file. Summary files allow users to identify which reference sequences receive a top score against the queried sequence. If a query sequence is derived from a species with many references present in the target database, many top results belonging to the same taxonomic group is expected. If the queried sequence is not strain-specific, broader homology presence in the target database can yield many top hits from related species, depending on the taxonomic representation within the database. In many cases, users may wish to explore non-maximum results. The parameter "–note_nmatch_p" specifies a threshold of matching bases, as a percentage of the entire query, above which *ATCCfinder* will report in summary tables. For example, if the value "0.95" is provided to *ATCCfinder*, all target homology matches to the query sequences occurring at or above 95% percent matching query base pairs are reported, regardless of top match status. *ATCCfinder* algorithms enable sequence-to-culturing automation by downloading purchasable ATCC sequences, aligning queries, and summarizing homologous alignments.

### ATCC genome retrieval

*ATCCfinder* tests implement searches against ATCC reference sequences, necessitating that this data be retrieved. *ATCCfinder* was used to download all available ATCC Genome Portal genome sequences on 22 January 2024. Since writing, hundreds of additional genomes have been added to the ATCC database, increasing the utility of *ATCCfinder*. ATCC genomes now require a paid subscription to access them, however a set of nine references are available for download for free. An vignette is included (Document S1) describing how to download references and perform an analysis using *ATCCfinder*.

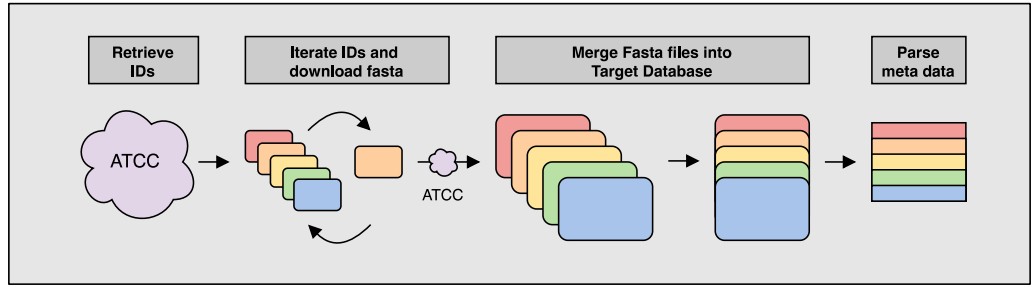

**Figure 1** **Overview of the *ATCCfinder* download algorithm** The ATCC API software package genome_portal_api (*The UniProt Consortium, 2021*) is used to identify all available genome IDs present on the ATCC Genome Portal (Retrieve IDs) followed by the iteration of individual IDs and downloading of reference sequences again using the ATCC API (Iterate IDs). Once all reference sequences have been downloaded, they are merged into a single multiline fasta file for use as the target database for searching query sequences against (Merge Fasta). Fasta headers are parsed for meta data information used in subsequent classification analysis (Parse meta data).

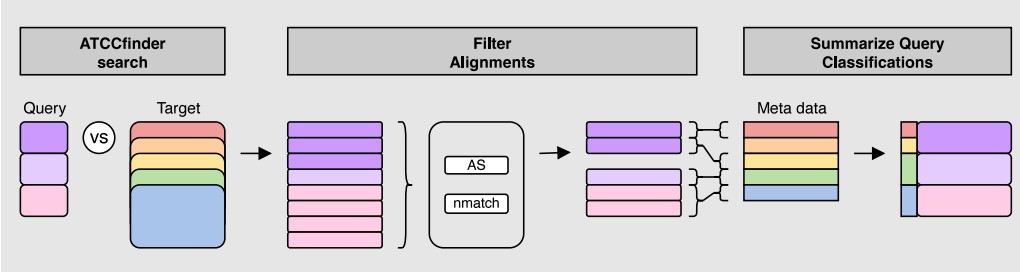

**Figure 2** *ATCCfinder* **search algorithm overview.** First, query fasta sequence(s) are aligned to the ATCC genome database sequences (ATCCfinder search). Next, top alignment results are identified by a summary metric of three statistics: maximum alignment score (AS), maximum number of matching bases (nmatch), and primary alignment type (tp) (Filter Alignments). Target IDs of filtered alignments are then linked with their associated taxonomic information using fasta-header-parsed ATCC meta data information followed by classification summarization by query sequenced (Summarize Query Classification).

## Algorithm validation using a known microbial community sample

The standardization, taxonomic diversity, and historical study of species present in the ZymoBiomics Microbial Community Standard (MCS) makes for an ideal set of taxa to test *ATCCfinder* alignment accuracy. Here, MCS is utilized as a sample with a known microbial community, with each constituent having high quality reference genomes available (Table 1). The MCS was used in three analyses (Fig. 3): (1) 16S/ITS amplicon sequences for MCS species were downloaded from the NCBI RefSeq database and queried. (2) All CDS sequences taken from MCS genomes on NCBI-RefSeq were queried. (3) 100, 500, and 1,000 base pair random sequences from MCS NCBI-RefSeq genomes were generated and queried using a custom R script. Random sequence generation was iterated 100 times, resulting in a collection of 100 sequences of three lengths for each species in the MCS. If classification results remained sufficiently ambiguous at 1,000 base pairs (analysis

**Table 1  RefSeq (amplicon) and GenBank (genome) accessions and statistics for ZymoBiomics Microbial Community Standard species used in** *ATCCfinder* **classification efficacy analyses.** Columns specify genus and species name (Taxonomy), RefSeq amplicon sequence accession (RefSeq amplicon), rRNA amplicon sequence type (rRNA), GenBank genome accession name (GenBank accession) and assembly version (GenBank Version), genome assembly size in megabases (Size (Mb)), genome assembly contiguous sequence quantity (Contigs), and genome assembly gene quantities queried (# CDS).

| Taxonomy | RefSeq amplicon | rRNA | GenBank accession | GenBank version | Size (Mb) | Contigs | # CDS |
|---|---|---|---|---|---|---|---|
| *Bacillus subtilis* | NR_112116.2 | 16s | GCA_000009045.1 | ASM904v1 | 4.2 | 1 | 4,325 |
| *Cryptococcus neoformans* | NR_171785.1 | ITS | GCA_000091045.1 | ASM9104v1 | 19.1 | 14 | 6,863 |
| *Enterococcus faecalis* | NR_115765.1 | 16s | GCA_000393015.1 | Ente_faec_T5_V1 | 2.9 | 16 | 2,680 |
| *Escherichia coli* | NR_114042.1 | 16s | GCA_000008865.2 | ASM886v2 | 5.6 | 3 | 5,289 |
| *Lactobacillus fermentum* | NR_104927.1 | 16s | GCA_029961225.1 | ASM2996122v1 | 2.1 | 2 | 2,072 |
| *Listeria monocytogenes* | NR_044823.1 | 16s | GCA_000196035.1 | ASM19603v1 | 2.9 | 1 | 2,855 |
| *Pseudomonas aeruginosa* | NR_114471.1 | 16s | GCA_000006765.1 | ASM676v1 | 6.3 | 1 | 5,571 |
| *Saccharomyces cerevisiae* | NR_111007.1 | ITS | GCA_000146045.2 | R64 | 12.1 | 16 | 6,001 |
| *Salmonella enterica* | NR_119108.1 | 16s | GCA_000006945.2 | ASM694v2 | 5 | 2 | 4,554 |
| *Staphylococcus aureus* | NR_118997.2 | 16s | GCA_000013425.1 | ASM1342v1 | 2.8 | 1 | 2,892 |

approach #3), then 100 additional sequences were generated and queried at lengths 1,500, 2,000, and 4,000 base pairs. (4) MCS sample DNA was used to generate Illumina sequencing data and the reads were queried with *ATCCfinder* (explained in detail in the next section (Fig. 3)). By testing a variety of sequence types (amplicon regions, gene sequences, randomly selected genomic sequences, short reads) from varied sources (sequence databases, lab-generated sequences) and wide taxonomic breadth (ZymoBiomics MCS species), the diversity of expected use cases should be captured. An additional analysis is included (Document S2) demonstrating *ATCCfinder* use with a curated strain-level data set.

## Microbial community DNA extraction, sequencing, & processing for algorithm validation with read-based sequences

Illumina sequences were generated to demonstrate *ATCCfinder*'s ability to handle read data, which could be used to link cultures to a reference isolates and associated metadata directly after sequencing. Briefly, 500 ng of ZymoBIOMICS Microbial Community DNA Standard (Cat. #D6306; Zymo Research) was fragmented using a Covaris E220 and Illumina library were prepared using NEBNext Ultra DNA II Library Preparation Kit with recommended protocols (Cat. #E7645L; New England Biolabs). Illumina libraries were eluted in DNA Elution Buffer (Cat. #D3004-4-10; Zymo Research). The concentration of the amplicon pools were obtained using the Qubit dsDNA HS Assay (Cat. #Q32854; Thermo Fisher Scientific, ). The average size of the library was determined by the Agilent High Sensitivity DNA Kit (Cat. #5067-4626; Agilent). An accurate library quantification was determined using the Library Quantification Kit—Illumina/Universal Kit (KAPA Biosystems, KK4824). The library was sequenced on the Illumina MiSeq generating paired-end 301 bp reads using approximately 43% of the MiSeq Reagent Kit v3 (600-cycle) (Cat. #MS-102-3003; Illumina). Illumina sequencing data was processed using EDGE (*Li et al., 2017*) to perform quality trimming and filtering (minimum Phred score of 20, minimum read length of 50 base pairs, and primer sequences were removed using AlignTrim). Reads were assembled into

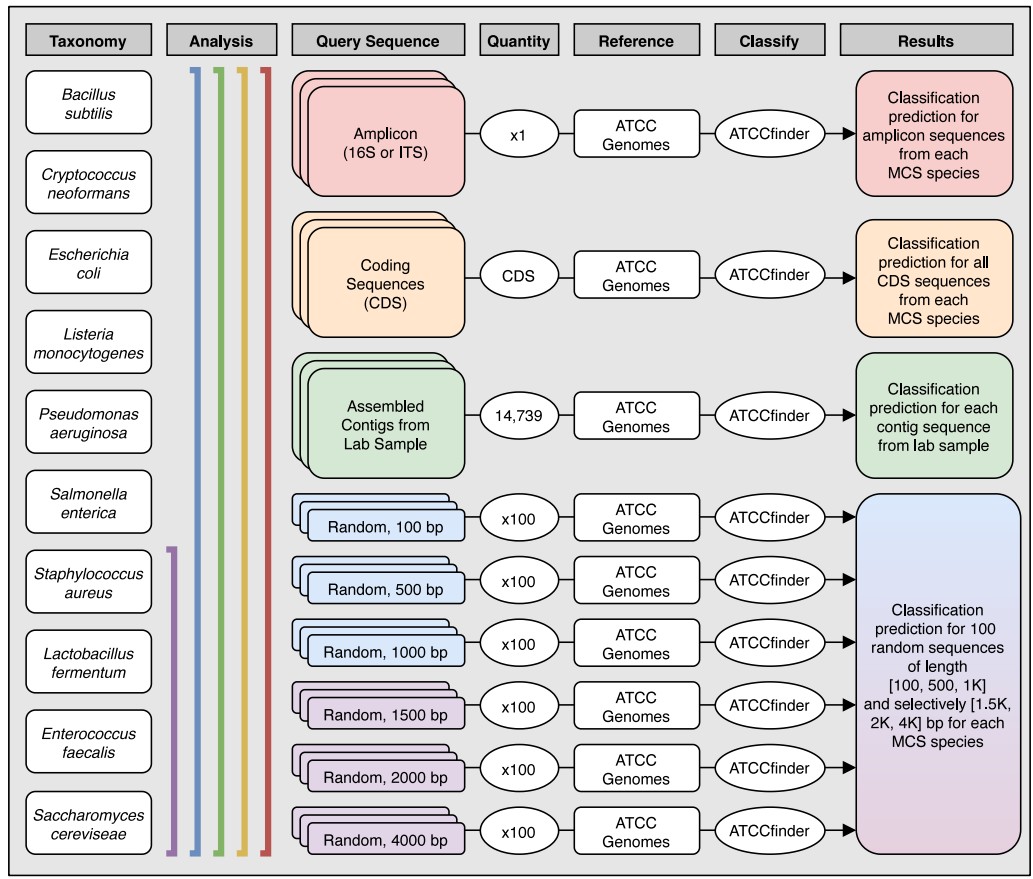

**Figure 3** **Analysis methods used to explore species level classification success of *ATCCfinder* using different types of sequences from a known microbial community sample.** Columns describe the species used (Taxonomy), the species included in each analysis (Analysis), what type of sequence was extracted and used as the experimental query to classify (Query Sequence), how many sequences were tested (Quantity), what target database the test query sequence was compared against (Reference) using the *ATCCfinder* classification tool (Classify), and the expected classification results (Results). The number of tested CDS sequences was unique for each species and corresponds to the total number of CDS present in each GenBank reference. See Table 3 for more details.

contigs using IDBA_UD (*Peng et al., 2012*) with kmers stepping from 31 to 121 in a step size of 20. BWA_mem (*Li, 2013*) was used to align reads to contigs for assembly validation. Contigs and raw reads were queried in *ATCCfinder* to identify candidate taxa.

# RESULTS

## ATCC reference genomes downloaded for target database

ATCC reference genomes were successfully downloaded on 22 January 2024 for use as targets to align query sequences against for homologous sequence identification. A total of 4,099 references were downloaded, representing 1,587 distinct taxa (Fig. S4) with total gzip compressed size of 8.7 GB. The majority of species have 1–3 genomes present (Fig. S4). 21 species have >0.5% relative database representation (21 or more genomes

**Table 2** **Top 30 most abundant species found in reference sequences downloaded from the ATCC genome database.** Columns describe the genus and species name (Taxonomy), the number of reference genomes present (n), and the relative abundance of these references out of the total 4,099 downloaded genome sequences (Relative abundance).

| Taxonomy | n | Relative abundance | Taxonomy | n | Relative abundance |
|---|---|---|---|---|---|
| *Escherichia coli* | 187 | 4.6% | *Human adenovirus* | 25 | 0.6% |
| *Salmonella enterica* | 150 | 3.7% | *Streptococcus agalactiae* | 25 | 0.6% |
| *Pseudomonas aeruginosa* | 94 | 2.3% | *Morganella morganii* | 24 | 0.6% |
| *Staphylococcus aureus* | 85 | 2.1% | *Streptococcus pyogenes* | 23 | 0.6% |
| *Klebsiella pneumoniae* | 82 | 2.0% | *Pseudomonas sp.* | 21 | 0.5% |
| *Streptococcus pneumoniae* | 71 | 1.7% | *Vibrio cholerae* | 21 | 0.5% |
| *Acinetobacter baumannii* | 55 | 1.3% | *Enterococcus faecium* | 20 | 0.5% |
| *Proteus mirabilis* | 42 | 1.0% | *Haemophilus influenzae* | 20 | 0.5% |
| *Campylobacter jejuni* | 38 | 0.9% | *Influenza B* | 20 | 0.5% |
| *Influenza A* | 37 | 0.9% | *Enterococcus faecalis* | 19 | 0.5% |
| *Listeria monocytogenes* | 36 | 0.9% | *Legionella pneumophila* | 19 | 0.5% |
| *Neisseria gonorrhoeae* | 34 | 0.8% | *Neisseria meningitidis* | 19 | 0.5% |
| *Yarrowia lipolytica* | 30 | 0.7% | *Pasteurella multocida* | 19 | 0.5% |
| *Serratia marcescens* | 28 | 0.7% | *Clostridioides difficile* | 17 | 0.4% |
| *Stenotrophomonas maltophilia* | 28 | 0.7% | *Shigella flexneri* | 17 | 0.4% |

present) of which eight were >1% (42 or more genomes present) (Table 2). A total of 1,055 (26%) species are represented by a single reference genome (Fig. S4). The composition of the downloaded ATCC genome database provides a diverse sequence set with variable taxonomic abundances for testing *ATCCfinder* against.

## Amplicon sequences successfully identify queried organisms

Amplicon sequences from ZymoBIOMICS MCS species were queried *via* a single *ATCCfinder* command against ATCC reference sequences. NCBI 16S and ITS amplicon sequences derived from Zymo MCS species ranging in size from 554–1,552 bp yield near-perfect identification results (Table 3). All species produce eight alignments total with 2–8 identified as *best* hits (Table 3, see Fig. 2 for 'best hit' criteria). The only imperfect result is *Bacillus subtilis* 16S amplicon sequences, yielding 1/5 best hits corresponding to *B. tequilensis* (Table 3). Commonly generated amplicon sequence data (16S or ITS) derived from distantly related taxa and ranging from 554–1,552 base pairs successfully identifies homologous reference sequences (Table 3), validating *ATCCfinder's* identification of commercially available strains for purchase from sequence data.

## Coding sequences promise high success rate identifying queried organisms

All coding sequences from each of the ZymoBIOMICS MCS species were queried *via* a single *ATCCfinder* command against ATCC reference sequences. Sequences from known microbial references are correctly identified 93.7% (*Saccharomyces cerevisiae*, 5,623/6,001 CDS) to 99.9% (*Staphylococcus aureus*, 2,891/2,892 CDS) at the species level (Fig. 4).

**Table 3 Species identification results of 16s and ITS amplicon sequences from a known microbial community sample.** Columns describe the source of the amplicon sequence (Taxonomy), the length of the amplicon sequence (Length), the maximum number of bases in a query that align to the target sequence (Max nmatch %), the total number of query hits against the entire downloaded ATCCfinder Genome Database (Hits Total), the number of hits with the highest cumulative score summed across maximum nmatch + primary alignment type + maximum alignment score (Hits Best), and the reported ATCCfinder classification result of references identified as being a "best hit" (Classification). Parenthesized Classification values indicate how many top hits contributed to each proceeding taxonomy classification out of the total Hits Best.

| Taxonomy | Length | Max nmatch % | Hits total | Hits best | Classification |
|---|---|---|---|---|---|
| *Bacillus subtilis* | 1,550 | 99.9% | 8 | 5 | (4/5) *Bacillus subtilis*, (1/5) *Bacillus tequilensis* |
| *Cryptococcus neoformans* | 554 | 98.0% | 8 | 3 | (3/3) *Cryptococcus neoformans* |
| *Enterococcus faecalis* | 1,483 | 100.0% | 8 | 6 | (6/6) *Enterococcus faecalis* |
| *Escherichia coli* | 1,467 | 99.8% | 8 | 2 | (2/2) *Escherichia coli* |
| *Lactobacillus fermentum* | 1,502 | 100.0% | 8 | 2 | (2/2) *Lactobacillus fermentum* |
| *Listeria monocytogenes* | 1,469 | 98.8% | 8 | 8 | (8/8) *Listeria monocytogenes* |
| *Pseudomonas paraeruginosa* | 1,489 | 99.9% | 8 | 8 | (8/8) *Pseudomonas paraeruginosa* |
| *Saccharomyces cerevisiae* | 752 | 99.9% | 8 | 3 | (3/3) *Saccharomyces cerevisiae* |
| *Salmonella enterica* | 1,539 | 99.4% | 8 | 2 | (2/2) *Salmonella enterica* |
| *Staphylococcus aureus* | 1,552 | 99.8% | 8 | 8 | (8/8) *Staphylococcus aureus* |

*Saccharomyces cerevisiae* also contains the largest percentage of sequences identified within the correct genus but incorrect species (6.3%, 378/6,001 CDS), *Escherichia coli* contains the largest percentage of sequences with wrong genus and species prediction (4.1%, 216/5,289 CDS), and *Lactobacillus fermentum* has the largest percentage of sequences without alignment hits (2.6%, 54/2,070 CDS) (Fig. 4). Many studies focus on the analysis of one or a few genes, so verifying species prediction capability using CDS sequences is an essential sequence type to validate *ATCCfinder* with. CDS sequences are diverse; sometimes remaining similar between closely related species while at other times rapidly diverging in sequence identity. In querying all CDS sequences from a known microbial community sample, we assess performance across a wide breadth of divergences. As with amplicon results, we see a high degree of strain identification success for queried CDS sequences across a range of 100's–1,000's of base pairs and diverse taxa (Fig. 4).

### *ATCCfinder* validation with random sequences

Randomly selected sequences from ZymoBIOMICS MCS species ranging in length from hundreds to thousands of base pairs were queried *via* a single *ATCCfinder* command against ATCC reference sequences. Using random sequences to identify strains is expected to perform poorly, as random genomic location sequences can include intergenic regions subject to genetic drift and are less likely to conserve evolutionary and phylogenetic signal. *ATCCfinder* performed well classifying randomly selected sequences. 9/10 species produce perfect or near-perfect species identification results, ranging between 65% (*Lactobacillus fermentum*) and 100% success (*C. neoformans*) (Fig. 5). *ATCCfinder* performed the worst at classifying *Lactobacillus*, where randomly selected 100 bp sequences have the lowest species-level success rate (65/100) (Fig. 5). The accuracy of randomly selected sequences improved with increasing sequence length. Higher successful identifications occurred at maximum tested lengths of either 1,000 bp or 4,000 bp for most queries (Fig. 5). The

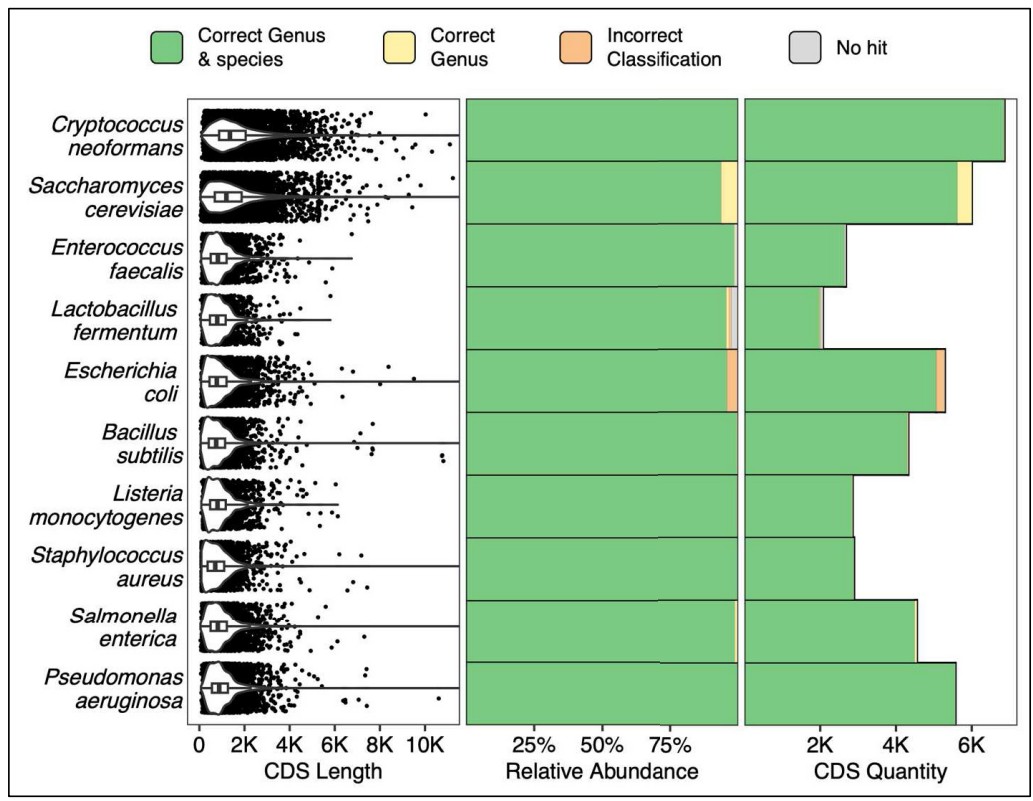

**Figure 4** **Classification results using all CDS (coding sequences) from a known microbial community sample.** The 10 $Y$-axis rows describe the queried MCS species while the $x$-axis columns describe coding sequence (CDS) length distributions (left), percent classification success (middle), and absolute classification success (right).

exception is *Lactobacillus fermentum*, where *ATCCfinder* consistently identifies sequences at the correct genus 94/100 (94%) times and at the correct species level 72/100 (72%) times, all independent of random sequence length (Fig. 5). *Lactobacillus* results stress the importance of database composition and curation awareness: when interpreting species identification results using *ATCCfinder* it is important to be aware of how many species-specific references exist in addition to measures of reference taxonomic and sequence accuracy. *Lactobacillus* recently underwent taxonomic reorganization, so extra caution should be exercised when interpreting results of species sequences within this clade. Like 16S, ITS, and CDS sequences, *ATCCfinder* performs strongly when identifying organisms from the more challenging random and/or non-coding DNA to identify commercially available strains closely related to queried taxa (Fig. 5).

## Classification of NGS reads and contigs from a known microbial community

Read sequences generated from a ZymoBIOMICS Microbial Community DNA Standard sample were queried against ATCC reference sequences using a single *ATCCfinder* command. *ATCCfinder* accurately identifies assembled contigs derived from sequencing

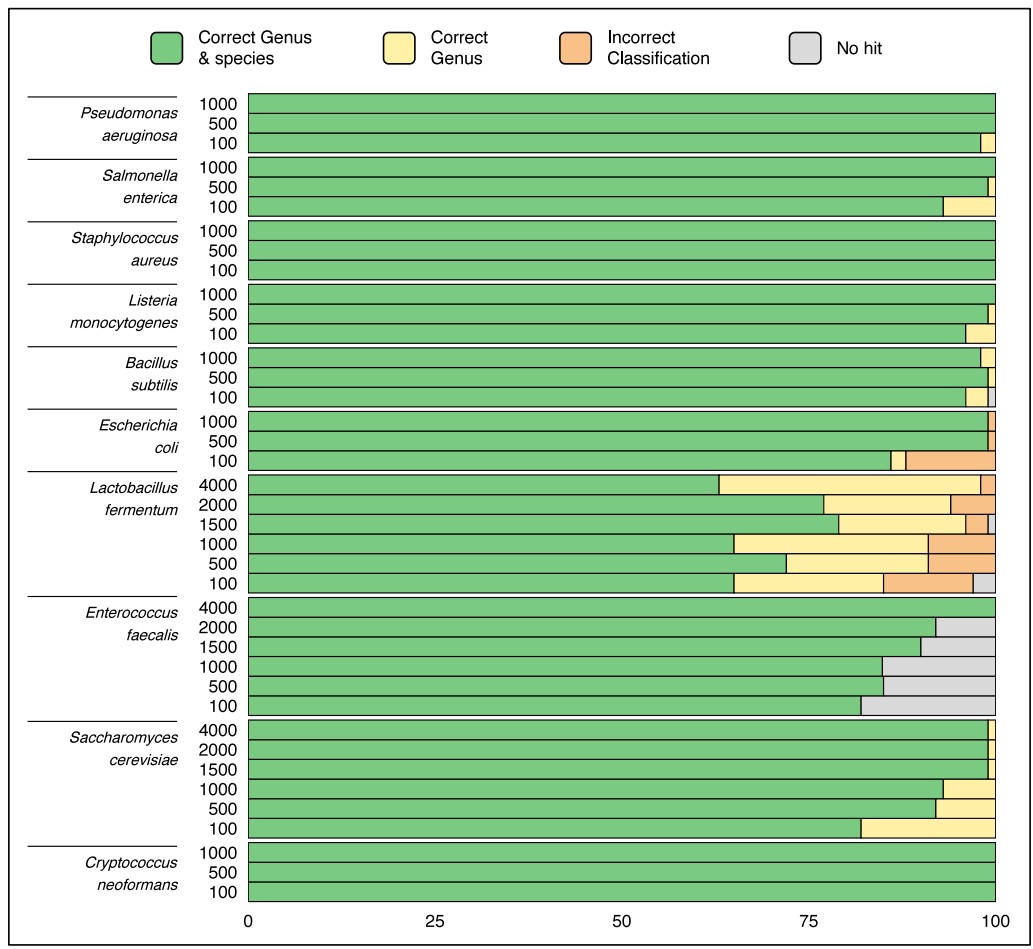

**Figure 5** **Classification results using 100 randomly sampled sequences of length 100–4,000 bp each from species in a known microbial community sample.** The X-axis describes what quantity out of 100 samples were successfully identified as the correct taxa while the Y-axis describes the length of sequences from the queried species. Species-level classification success improves as random sequence length increases in all tested species except for *Lactobacillus fermentum*.

a known microbial community sample. Sequence assembly resulted in 14,7639 contigs, ranging in size from 200 to 468,709 bp ($N_{50}$ = 15,860 bp, mean = 4,292 bp, median = 958 bp). Of these sequences, 94.6% produced homology results belonging uniquely to one of the ten species present in the sequenced sample (Fig. 6). For correct species-classified sequences, taxonomy assignments reflect expected proportions based on complete genome sizes ($R^2$ = 0.855) (Fig. S1).

## DISCUSSION

Here we demonstrate *ATCCfinder*'s ability to handle a variety of sequence types and lengths to correctly identify organisms available from ATCC which carry the sequence of interest. Regarding species-level identification, *ATCCfinder* is successful regardless of whether the reference database contains few (*Lactobacillus fermentum*, *C. neoformans*) or
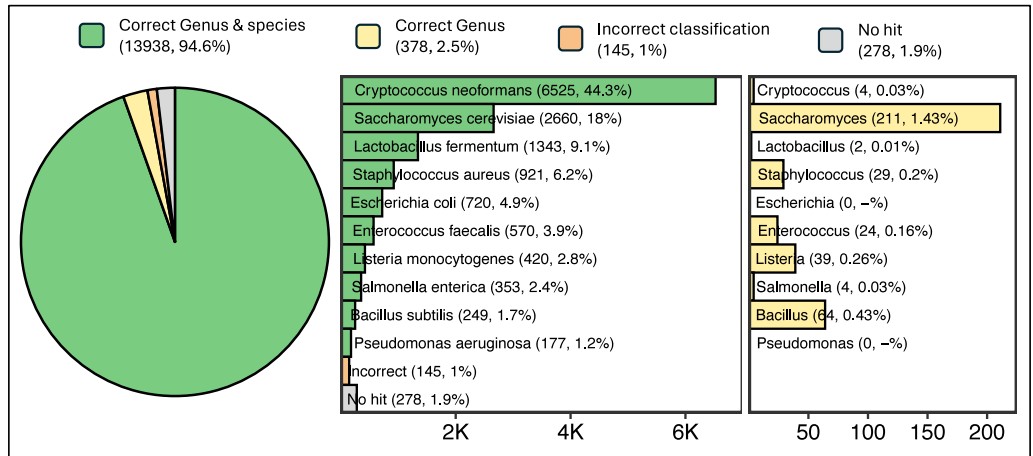

**Figure 6 Results of classification of 14,739 contigs sequenced and assembled from a known microbial community sample.** Parenthetical integers describe the total quantity of contigs in each category, with percentage values describing integer abundance relative to the total quantity of contigs analyzed (14,739).

tens (*Escherichia coli*, *Salmonella enterica*) of species-specific genomes. While *ATCCfinder* performs strongly, highly unique queries, which represent a species that does not have a close relative in the ATCC catalogue, may not return meaningful results. With any number of correct matches, it is always recommended that users consult the "n_match_%" column in the final report to gauge the quality of the match, as the best hit may align well to many similar references, but all at 50% of the total query length, suggesting possible misidentification or the absence of a closely related species in the reference database. "Maximum" values for top-alignment statistics may be misleading if the alignment with the top aggregate score did not achieve top marks across all considered scoring statistics (*i.e.,* events where the cumulative score is less than 3). When considering a result, consult the annotated pairwise mapping format file for details on individual alignments and what specific statistics contributed to their aggregate score and utilize the "−note_nmatch_p" parameter to report sub-maximum homology. Regarding queried sequence length, *ATCCfinder* performs better as length increases, although sequences of 100 bp are frequently classified correctly to the species level. Species prediction success does not correlate with CDS genome density (Table S1) or the quantity of features present (Fig. S3), suggesting that success of random sequence classification does not rely upon the presence of more-conserved sequence types or gene-related motifs. The number of database references that belong to the same species, the same genus, or closely related genera may affect the likelihood of successful strain identification. At the time of analysis, the ATCC genome database contained five *Lactobacillus fermentum* genomes with 29 other references belonging to 12 other species within the genus. When testing the success of random sequences, we observe many *Lactobacillus fermentum* queries being identified as the correct genus but incorrect species. The database contains 187 *Escherichia coli* and 17 *Shigella flexneri* references (Table 2) that may explain the small proportion of queried *Escherichia coli* CDS sequences being identified as the incorrect genus and species (Fig. 4).

*ATCCfinder* identification ability is validated with a variety of sequence data types (16S and ITS amplicon, coding sequences, random genome sequences), suggesting that the tool may succeed regardless of the kind of queried sequence data type. Amplicon sequences, which are highly conserved due to essential ribosomal function, are frequently used to classify broad taxonomic groups in metagenomic samples but often fail to classify to the species level (*Golob et al., 2017*). *ATCCfinder* is demonstrated to successfully query and interpret full-length ribosomal sequences, reporting species-level identification while avoiding feasible but ultimately incorrect results from other closely related species within the same genus that may contain very similar target sequences. Coding sequences, comprised of nucleic acids translated into proteins, are long queries that perform very well within the tool to produce correct species homology predictions. *Escherichia coli* had the poorest CDS performance with 4.1% sequences not correctly being identified as *Escherichia*, instead being predicted as within the genus *Shigella,* which is a known to render Escherichia polyphyletic (*Beld & Reubsaet, 2012*). The most problematic tested sequence type is random genome samplings, especially those of shorter lengths (100 bp). When performing random genome sampling, there is a likelihood of selecting intergenic or otherwise less informative regions of the genome that may be less characterized and therefore harder to produce unambiguous homology results from. Random sequences which capture CDSs are likely more conserved than non-coding sequence and could thus enhance *ATCCfinder's* identification success. Interestingly, no relationship exists between correct species identification of a random sequence and how many features were present on the query (Fig. S3). It is important to note the lack of consistency inherent in this random sampling exercise, as NCBI reference genomes differ in their source and quality. Despite these likely underlying differences, we see similar feature quantity patterns in sampled genomes, with fungal assemblies consistently including more features than their bacterial counterparts (Fig. S2), suggesting that clade may be a stronger database characteristic than reference quality itself. While identification success increased for most species as the query sequence length was increased to 1,000 or 4,000 base pairs, *Lactobacillus fermentum* remained consistently mediocre. *Lactobacillus fermentum* has a CDS density similar to other tested species (Table S1), as well as a species-to-genus presence ratio in the ATCC database similar to that of *B. subtilis*. Notable outlier characteristics of *Lactobacillus fermentum* are that it has the smallest tested genome (2.1 Mbp, Table 3) and contains the least reference genomes in the target database (5) although *C. neoformans* and *B. subtilis* are close with seven and 10 references respectively. Across all 600 randomly sampled sequences, 70% are correctly identified to be *Lactobacillus fermentum* while other common incorrect identifications are *Lactobacillus paracasei* (14%), *Lactiplantibacillus plantarum* (5%), *Lactobacillus gasseri* (3%), *Lactobacillus helveticus* (3%) which may be caused by ATCC species misclassification or horizontal gene transfers. It is notable to observe that amplicon and CDS sequences do not follow this outlier pattern in *Lactobacillus fermentum*, instead producing near-perfect species identifications across all CDS the amplicon sequences derived from the same reference assembly, a result consistent with all other tested species. The quantity of features present on random sequence queries is also not abnormal, being quite similar in profile to *Enterococcus faecalis* (Fig. S2) which yielded better identification results as random sequence query length increased (Fig. 5). If

a researcher is interested in species closely related to *Lactobacillus* or species with small and compact genomes similar to *Lactobacillus*, the performance of *ATCCfinder* may suffer due to characteristics described here. The results of *Lactobacillus fermentum* demonstrate that close consideration of alignment results is necessary when the classified query is associated with genera less represented in the target database. In such instances, additional manual validation *via* other tools such as BLAST (*Camacho et al., 2009*) searching against NCBI-nt or Ref-seq databases is recommended.

Performed analyses are with 1–15,000 sequences ranging in size from 100's to 100,000's of bases in size. These quantity and size ranges are the recommended limitations of the software, which are additionally biologically relevant, sequences too much shorter are likely uninformative and sequences too much longer are increasingly easier to match to known organisms or have originated from known organisms. Importantly, the upper bounds of number and length of sequences remain executable on a modern MacBook computer (3.1 GHz Quad Core Intel Core i7, 16 GB RAM) in a few hours, making even extensive searches feasible. Running 1–100 queries of ~1 Kbp can be performed in minutes.

## ACKNOWLEDGEMENTS

This program was produced under U.S. Government contract 89233218CNA000001 for Los Alamos National Laboratory (LANL), which is operated by Triad National Security, LLC for the US Department of Energy/National Nuclear Security Administration. All rights in the program are reserved by Triad National Security, LLC, and the US Department of Energy/National Nuclear Security Administration. This work has been cleared for public release by Los Alamos National Laboratory (LA-UR-24-25680).

### Funding

This work was funded by a Los Alamos National Lab Early Career Research Award through Laboratory Directed Research and Development (20220554ECR) to Erik R. Hanschen. The funders had no role in study design, data collection and analysis, decision to publish, or preparation of the manuscript.

### Grant Disclosures

The following grant information was disclosed by the authors:
Laboratory Directed Research and Development: 20220554ECR.

### Competing Interests

The authors declare no potential financial or other interests that could be perceived to influence the outcomes of the research.

The authors declare there are no competing interests.

## Author Contributions

- Samuel I. Koehler conceived and designed the experiments, performed the experiments, analyzed the data, prepared figures and/or tables, authored or reviewed drafts of the article, and approved the final draft.
- Earl A. Middlebrook conceived and designed the experiments, analyzed the data, authored or reviewed drafts of the article, and approved the final draft.
- Blake T. Hovde conceived and designed the experiments, analyzed the data, authored or reviewed drafts of the article, and approved the final draft.
- Erik R. Hanschen conceived and designed the experiments, analyzed the data, authored or reviewed drafts of the article, and approved the final draft.

## DNA Deposition

The following information was supplied regarding the deposition of DNA sequences:

The sequences are available at NCBI SRA: SRR33795049,

## Data Availability

The software is available at GitHub and Zenodo:

– https://github.com/lanl/ATCCfinder.
– Koehler. (2025). ATCCfinder. Zenodo. https://doi.org/10.5281/zenodo.15178103.

## Supplemental Information

Supplemental information for this article can be found online at http://dx.doi.org/10.7717/peerj.19832#supplemental-information.

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
