# Peer review of "Use *ATCCfinder* to identify commercially available American Type Culture Collection strains based on sequence queries"

_PeerJ, doi:10.7717/peerj.19832_

## Round 0.1 · original submission · Major Revisions

Dear authors, thank you for submitting your manuscript for consideration by PeerJ. Three experts in the field have carefully assessed your study and recommend revising your work. Please go through your manuscript and make a thorough improvement to meet the journal's and reviewers' criteria. A point-by-point response letter is also required to clearly show that you have responded to all the comments and concerns when resubmitting your manuscript.

Reviewer 1 ·

Basic reporting

The manuscript is written in clear English. Motivation for the tool is well stated in the introduction, though specific examples of other tools/resources are not cited, e.g. "While some databases avoid requiring local download by directly offering sequence searching capability built into their web interfaces" which presumably refers to the ATCC itself. However, literature is appropriately referenced throughout the remainder of the manuscript.

The manuscript structure is sound, figures and tables are well described and easy to understand.
Software is open source and freely available on Github (https://github.com/lanl/ATCCfinder/) and raw sequencing reads were also made available via Google drive.

Experimental design

As stated above, the research question is well defined though it could use some specific examples of other tools/resources which do not accomplish what ATCCfinder does. However, I can see the value add of a wrapper specifically for the ATCC database which can automate the process of database creation to purchasable strain identification.

The choice of benchmarks is well rationalised and benchmarks are comprehensively performed, demonstrating ATCCfinder's ability to effectively query with multiple sequence types. Limitations of the identification power of ATCCfinder for certain strains (i.e. Lactobacillus) is also well noted in the discussion. The lack of any comparison to the ATCC webservers built in search functionality is a notable omission. While I appreciate it has significantly less throughput than ATCCfinder (and thus the benchmark is not applicable), I imagine it is still sufficient for users who are simply wanting to find a strain for a given nucleotide sequence. I would be interested to see if the Lactobacillus strains are correctly identified using the webserver for example.

Methods are sufficiently described, though there could be more detail regarding the identification of 16S/ITS/CDS sequences. The GitHub repository (link is not actually shown in the manuscript main text) contains instructions on installation and use of the tool which are easy to follow.

Validity of the findings

As I noted in email correspondence to the editorial desk, however, my biggest concern is that as of May 2024, the ATCC genome database has now been placed behind a paywall (~$1800 a year for individuals as per https://genomes.atcc.org/plans). The authors retrieved the data used in the manuscript prior to this date, presumably when the genomic data was freely accessible. I would also imagine this guided the authors decision to develop the software in the first place. As I do not have a subscription to this catalogue, I am unable to test the software whatsoever - I simply get an error when attempting to use the download script.

While I can imagine the software could still be useful to users with a subscription, I am not so comfortable recommending it given I cannot test it personally and cannot reproduce any of the results. Moreoever, given you must now pay just for access to the catalogue, it would seem that the motivation for the tool (i.e. being able to identify ATCC strains prior to purchase) has been weakened as a result.

·

Basic reporting

Structure of the Document
- The paper adheres to the main sections required by PeerJ. However, the subsections in the "Materials and Methods" and "Results" sections do not follow the formatting guidelines specified in the “Author Instructions.” Specifically, subheadings should be bold, followed by a period, and start a new paragraph (e.g., Background. The background section text goes here...).
- The Acknowledgements section currently includes funding information, which should be relocated to the appropriate section as per the “Author Instructions.”
Overall, the paper would benefit from more precise wording to enhance clarity for the reader.

Abstract
- The first sentence could be strengthened by being more specific about the extent of the research. Instead of beginning with "Much," consider providing quantifiable details to underscore the importance of the topic. For example, reference the frequency of this type of query to establish stronger stakes.
- While ATCC is an excellent resource, its role isn't introduced as clearly as it could be. It would be beneficial to explain its significance relative to other databases like NCBI. For instance, mentioning that ATCC houses over 4,000 reference genomes could highlight its importance.
- Clarify whether all ATCC data is included in larger databases like NCBI, and explain how likely one is to find a strain in ATCC compared to other repositories.
- Discuss whether there are alternative methods for querying ATCC data, such as using the BLAST interface, and compare these to the functionality provided by ATCCFinder.
- The conclusion of the abstract should more explicitly tie into the stakes established earlier, such as by stating, "ATCCFinder enables users to identify historical strains in the ATCC database using known functional genes, operons, other genetic components, or homologous sequences."

Introduction
- The introduction would benefit from incorporating references to previous research and efforts that have addressed similar problems. Currently, there are no citations, which leaves a gap in the contextual foundation of the paper.
- It would be helpful to discuss what researchers currently do to find their strains, rather than just what they need to do.
- Consider addressing why this functionality isn’t already provided by ATCC and whether it should be integrated. Additionally, compare ATCCFinder to similar tools available in other resources like NCBI.

Methods and Materials
- A brief introductory paragraph would be helpful to explain how the subsections of this section fit together.
- Include links to the ATCCFinder GitHub repository and another resource like Zenodo. Ensure that the GitHub repo has a tagged release used for this publication, enabling replication of the work. This tagged version could also be archived in Zenodo along with all necessary files and code.
- While minimap2 was likely a suitable choice, an explanation of why it was chosen over other tools like BLAST, particularly since BLAST is provided by ATCC, would strengthen the section. A comparison between the two would be valuable.
- Consider Dockerizing the tool so users can run a prebuilt container, ensuring it can be provisioned and executed as designed.
- Organize the section so that steps are presented in the order they would be performed, e.g., genome retrieval before running the algorithm.
- The generation of the set of contigs and querying them with ATCCFinder is a good test, but it should be tied back to the stakes outlined earlier. Explain why someone would want to find strains based on newly generated contigs using ATCCFinder.
- Introductions and concluding sentences should be added to each subsection for continuity. A concluding paragraph that ties the subsections together would also improve the flow.

Results
- Like the Methods section, the Results section would benefit from introductory and concluding sentences that provide continuity.
- Subsections should closely align with those in the Methods section.
- It would be beneficial to explicitly state how many species are in the dataset. A figure showing the distribution of genomes per species would help the reader understand the genetic relationships within the dataset.
- The results should more thoroughly explain the information presented in the tables and figures. There is some confusion as Table 3 is referenced in both the Methods and Results sections, making it difficult to distinguish between methods and findings.
- Rephrasing sentences to avoid ending with prepositions would enhance the formal tone of the paper.
Figure 4, indicates the number of species/genus classifications that were correct. Analyzing the incorrect classifications to determine if the match quality or another metric could predict false classifications would be useful.
- Clarify how hits in multiple species for a single contig are handled. Are only the top hits considered? If a second hit is incorrect, how is this reflected?
- Change “produced incorrect genus and species predictions ~6/100 times” to “produced correct genus and species predictions 94% of the time” to emphasize the tool's high accuracy. This point should be more prominent as it is central to the paper's value.

Discussion
- Consider discussing the potential for integrating this tool into the ATCC resource itself. Mention that running this tool locally may be challenging for some users due to the need to download large datasets. - Suggest the possibility of hosting a service to reduce barriers.
- The idea of providing a service could be explored further, particularly if there are plans to host ATCCFinder as a service for users.

Experimental design

The experimental design is sound and demonstrates that the tool often accurately identifies the species/genus of a sequence. However, the paper should better justify why users would need this tool, given that genus and species are often already known in modern experiments. The paper should address whether the use case of identifying strains from older studies without reference genomes is significant enough to warrant a separate tool. Perhaps the experimental design could point out sequences in old papers where it does not state the genus/species and see how often it can identify a match with high confidence.

Validity of the findings

The findings are valid but should be compared to results obtained from alternative techniques, such as BLAST. Since minimap2 is designed for this type of search, it should outperform BLAST, and this difference should be quantified.

Additional comments

The paper shows a lot of promise and has the potential to make a significant contribution to the field. With some revisions, particularly in clarifying the importance of the tool, addressing formatting issues, and ensuring the Methods and Results sections are distinguished, it could be a strong candidate for publication.

Reviewer 3 ·

Basic reporting

Introduction completely lacks of literature background and references (even to the main database at stake: 10.1128/MRA.00818-21). Please provide reference to the software, databases, and facts mentioned (what are the “standard workflows” mentioned at L1? Any reference?).

Experimental design

In my opinion, in the present form, the software fails at answering the question asked in the introduction “ Did the database queried against not produce any convincing alignments?”, and only proposes a very slight improvement compared to a manual curation.
Indeed, the automation is only partial as a large fraction of the work to be done to select the right strain has to be done “at hand” (I am referring to the comments in the discussion). Maybe some of the advises given in the paper on the interpretation of the results could be embedded in the present software (flag unsure decision). The current automation seems to roughly replace a standard homology search, which is in my opinion not a heavy step to carry out by hand.

Moreover, the evaluation of the performances has mainly been done using a sample with very low complexity (not encompassing a mixture of different strains of the same species). Controlled in sillico evaluation using advanced software such as (https://microbiomejournal.biomedcentral.com/articles/10.1186/s40168-019-0633-6) with closely related strains as input, should be carried out to be convincing.

Validity of the findings

I could not find not test the presented software. I am not aware of its availability, openness nor usability. It is crucial to mention it in the current manuscript.

Additional comments

Minor:
L174: The choice of IDBA_UD is surprising as it is not a well performing assembler (see https://doi.org/10.1093/gigascience/giac122, https://academic.oup.com/bib/article/24/2/bbad087/7077274), although it does not impact much the study here.

Typo/vocabulary:
L88: Did the database queried against not produce any convincing alignments? → please use a positive instead of negation question
L114: “random” sequences should be better described as sequences extracted from randomly selected location on the genome. Otherwise, it looks like randomly generated sequences.
L141: I guess the author wants to refer to Table 1 instead of Table 3.
L149: I guess the author wants to refer to Table 2 instead of Table 3.

---

## Round 0.2 · Minor Revisions

Please revise the manuscript based on the reviewer's comments

·

Basic reporting

Grammar:

"can a complicated task" -> "can be a complicated task"

"percent identify" -> "percent identity"

Experimental design

looks good

Validity of the findings

looks good

Additional comments

The Article has been much improved!!!! I believe you have addressed all of my concerns. All that is left is the fixing of the minor grammatical issues.

---

## Round 0.3 · accepted · Accept

The manuscript can be accepted for publication in its current form.